# FEIN-Z: Autoregressive Behavior Cloning for Speech-Driven Gesture Generation

Leon Harz*
lharz@techfak.uni-bielefeld.de
Bielefeld University
Germany

Hendric Voß*
hvoss@techfak.uni-bielefeld.de
Social Cognitive Systems Group
Bielefeld University
Germany

Stefan Kopp
skopp@techfak.uni-bielefeld.de
Social Cognitive Systems Group
Bielefeld University
Germany

## ABSTRACT

Human communication relies on multiple modalities such as verbal expressions, facial cues, and bodily gestures. Developing computational approaches to process and generate these multimodal signals is critical for seamless human-agent interaction. A particular challenge is the generation of co-speech gestures due to the large variability and number of gestures that can accompany a verbal utterance, leading to a one-to-many mapping problem. This paper presents an approach based on a Feature Extraction Infusion Network (FEIN-Z) that adopts insights from robot imitation learning and applies them to co-speech gesture generation. Building on the BC-Z architecture, our framework combines transformer architectures and Wasserstein generative adversarial networks. We describe the FEIN-Z methodology and evaluation results obtained within the GENEA Challenge 2023, demonstrating good results and significant improvements in human-likeness over the GENEA baseline. We discuss potential areas for improvement, such as refining input segmentation, employing more fine-grained control networks, and exploring alternative inference methods.

## CCS CONCEPTS

• **Human-centered computing** → **Interactive systems and tools**; *Empirical studies in interaction design*; **HCI theory, concepts and models**; • **Computing methodologies** → **Neural networks**; **Learning latent representations**; *Unsupervised learning*.

## KEYWORDS

machine learning; deep learning; co-speech gesture generation; gesture synthesis; multimodal data; transformer; behavior cloning; reinforcement learning

**ACM Reference Format:**
Leon Harz*, Hendric Voß*, and Stefan Kopp. 2023. FEIN-Z: Autoregressive Behavior Cloning for Speech-Driven Gesture Generation. In *INTERNATIONAL CONFERENCE ON MULTIMODAL INTERACTION (ICMI '23), October 9–13, 2023, Paris, France.* ACM, New York, NY, USA, 10 pages. https://doi.org/10.1145/3577190.3616115

---

*Both authors contributed equally to the paper

## 1 INTRODUCTION

Human communication is a multifaceted process that relies on various modalities, including verbal expressions, facial cues, and bodily gestures. Combining these modalities allows us to convey complex messages and facilitate meaningful interactions [9, 50]. Consequently, the development of machines that can process and generate these multi-modal signals is crucial to enable seamless interaction between humans and agents. A key aspect that makes gesture generation particularly challenging is the existence of multiple valid gestures for a given interaction. Unlike verbal expressions, which often have a single intended meaning, gestures can convey different nuances and interpretations, leading to a one-to-many mapping problem [41]. Capturing this inherent variability and generating contextually appropriate gestures is a complex task that requires careful consideration. The importance of gesture generation extends beyond research to practical applications in real-world scenarios and virtual environments. In human-robot interaction, gestures play a crucial role in enhancing communication and facilitating natural interactions between humans and robotic agents [56]. Similarly, in virtual reality, realistic and expressive gestures contribute to immersion and engagement, enabling more intuitive and compelling experiences [35]. Therefore, the development of robust and effective gesture-generation methods has great potential for improving various areas of human-machine interaction.

In this work, we propose the FEIN-Z framework, a combination of the proposed *Feature Extraction Infusion Network (FEIN)* and the zero-shot learning aspect of the BC-Z architecture *(Z)*. Inspired by recent achievements in robotic imitation learning, we extend the BC-Z approach [27] intended to generalize robotic manipulation tasks to unseen problems, to the co-speech gesture generation domain. As transformer architectures have shown promising results in a wide variety of domains [17, 48], including co-speech gesture generation [38], we replace and extend multiple components of the original BC-Z approach with a transformer architecture. Generative adversarial networks (GAN) are widely used in the robotic and co-speech gesture generation domain [20, 52]. Building upon the insight gained from recent approaches [52], we propose to use a Wasserstein generative adversarial networks (WGAN) with a Wasserstein divergence objective to guide our framework to generate natural and expressive gestures. The released evaluation results of the GENEA Challenge 2023 show that our framework outperforms the challenge baseline with regard to human-likeness by a significant margin and ranks in the top half of all evaluated approaches [31]. In the next sections, we will first give a brief overview of the existing work and current achievements of co-speech gesture

generation (Section 2), before detailing the proposed *FEIN-Z* architecture, the individual components, the data processing, and our training procedure (Section 3). Finally, we will discuss the results of the performed evaluation (Section 4) and conclude with an outlook for possible improvements of our work (Section 6).

## 2 RELATED WORK

Gesture generation is an area of research that is rapidly progressing. Previous studies have explored various approaches, initially focusing on rule-based methods [10, 29, 34, 40] and simple computational models [8, 19], and later transitioning to early machine learning techniques [12, 23]. Currently, data-driven approaches that integrate multiple modalities are being employed [4, 41, 59], advancing the field even further.

Initially, gesture generation relied on manually crafted rules, either directly applied to specific avatars or used in conjunction with computational models that estimated appropriate gestures based on accompanying speech [10, 19, 29, 34]. Although these approaches generally struggled to produce natural and fluent gestures, they did enable the creation of complex representative gestures that are challenging to achieve with current data-driven methods [5, 6, 29, 34].

During the beginning of data-driven gesture generation, the focus was primarily on single modalities, where gestures were generated based on previous gesture frames [47], textual inputs [12, 56], or audio-driven inputs [18, 21, 23]. Recent research has witnessed a notable shift towards the generation of multi-modal co-speech gestures. This approach integrates gestures with audio, text, and other input modalities to produce varied and natural gestures. To accomplish this, advanced techniques such as general adversarial networks (GANs) [3, 41, 52, 54, 55], cyclic functions [26], glow networks with invertible convolutions [24], variational autoencoders [38, 46], and deep reinforcement learning have been used [46]. Recurrent neural networks, specifically Bi-Directional Long Short-Term Memory (Bi-Directional LSTM) and gated recurrent unit (GRU) [13, 25], have demonstrated the ability to generate natural co-speech gestures [23, 57], with various adaptations of recurrent architectures still being utilized in recent approaches [28, 30, 44, 51]. Notably, the incorporation of style embeddings has facilitated the generation of distinct gesture styles for individual speakers, thereby enabling diverse variations in gestures that are tailored to specific styles or speakers [21, 55].

Recent advancements in the field of co-speech gesture generation can be broadly categorized into two main approaches: retrieval-based methods and learning-based methods. Retrieval-based methods involve the creation or learning of predefined sets of gesture units and employ techniques such as keyword matching, semantic analysis, and prosody analysis to retrieve corresponding gestures from a comprehensive database [59]. Conversely, learning-based methods focus on training models to directly predict co-speech gestures using paired co-speech gesture data [55]. In recent studies, some researchers have automated the creation of gesture unit databases by leveraging training data. These gesture units are then employed to train deep learning models, enabling the generation of new and varied co-speech gestures [38]. Both retrieval-based

and learning-based methods have proven to be effective in addressing the inherent challenge of one-to-many mapping in co-speech gestures [11, 32, 44, 55]. Notably, recent work on retrieval-based methods have even demonstrated superior performance compared to ground truth gestures [58, 59].

Simultaneously, significant progress has been made in the realm of reinforcement learning for robot control, particularly in the utilization of text and visual data as input. Within this context, text data is commonly employed either as action descriptions or goal descriptions. Recently, successful approaches have emerged leveraging large language models (LLMs), which generate suitable plans for given goals [1] [42] [36]. These approaches harness LLMs to break down goal descriptions into a sequence of feasible low-level actions expressed in natural language. Subsequently, the action descriptions undergo embedding and serve as additional input to a reinforcement learning model. As an example, PaLM-SayCan incorporates the BC-Z network [27] to acquire low-level robot skills by providing visual data of the current state alongside text descriptions of planned actions.

Both the co-speech gesture generation and reinforcement imitation learning domains share a common goal: to generate elaborate and complex outputs by acquiring knowledge from a relatively limited data set. As the imitation learning domain has made significant progress in minimizing the data requirements for generating complex outputs, we believe that these achievements can be leveraged in the gesture generation domain. Therefore, we propose our novel framework, which is built on the foundation of imitation learning, with the expectation of extending these advances to gesture generation.

## 3 MODEL AND METHOD

Our framework builds upon the BC-Z architecture by Jang et al. [27], which is a flexible imitation learning system that can learn from both demonstrations and interventions for a given Zero-Shot task. Similar to our approach, the BC-Z architecture generates its output in an autoregressive manner. However, given the unique domain and data characteristics of co-speech gestures, we have made several modifications to the backbone of the BC-Z architecture to adapt it to our domain. In particular, we replaced the vision network component of BC-Z with an attention-based network that takes inputs from each modality (*Transformer Network*). In addition, we refined the Feature-wise Linear Modulation (FiLM) network [43], while retaining the fundamental concept of linear modulation applied to the previous embedding. We refer to this modified FiLM architecture as the *Feature Extraction Infusion Network (FEIN)*. Our framework takes audio, text, and speaker identity information from both the main agent and the interlocutor as input, alongside gestures from the interlocutor. To incorporate the temporal dimension of the provided data, we employ positional encoding techniques proposed by Vaswani et al. [49]. The transformer network receives audio features, text features, and speaker identity information from both the main agent and the interlocutor. The FEIN module also utilizes this data, with the addition of previous $t$-gestures from both the main agent and the interlocutor. The output of the transformer network is then combined with features extracted from the FEIN module. The resulting embedding is further processed by a

joint-specific Fully Connected Network (FCN). In addition to the architectural refinements, we utilize a Wasserstein GAN network with gradient divergence (WGAN-div) to improve the generation performance of our framework [53]. To enhance the generation performance of our framework we employ a discriminator with an FCN consisting of four linear layers, using the leaky ReLU activation function [39]. Figure 1 gives an overview of our approach. In the following sections, we will provide a detailed description of the sub-modules of this framework, including the attention-based network, FEIN, and the control network.

## 3.1 Transformer Blocks

The presented framework incorporates a total of four transformer blocks, each possessing a consistent underlying architecture with distinct parameters. These blocks comprise a multi-attention head followed by a feedforward network. To augment the capabilities of the feedforward network, we have introduced the Swish-Gated Linear Unit (SwiGLU) activation function [45] into the transformer blocks. As a result, the output $y$ of the transformer blocks can be computed as follows:

$$\text{MultiHead}(Q, K, V) = \text{Concat}(\text{head}_1, \ldots, \text{head}_n)W^0 = x \quad (1)$$

$$f(x) = \text{Swish}(x \cdot W_1) \otimes (x \cdot W_2) \quad (2)$$

$$y = f(x) \cdot W_3 \quad (3)$$

In the above equations, MultiHead denotes the multi-headed attention layer, Swish represents the swish activation function and $W$ corresponds to the weights of the linear functions.

## 3.2 Transformer Network

The BC-Z framework initially relied on visual data, specifically images, to predict robot actions based on the current context. However, our specific scenario lacks visual data, therefore requiring modifications to the original architecture. To address this challenge, we adopt a transformer network, known for its capacity to model long-term dependencies within structured input data. Central to our approach is the integration of audio and text input from both the main agent and the interlocutor. Particularly, audio and text data are processed independently. For each input modality, the framework computes an attention-based embedding, which learns the information and relationships present within the data. The individual attention-based embeddings obtained in the preceding step are then aggregated and passed through an additional multi-attention mechanism, known as the 'Combined Transformer'. This combination stage aims to identify and encapsulate important cues related to the interplay between audio and text data. The resultant composite embedding effectively captures salient information and data relationships, forming the fundamental basis for subsequent processes.

## 3.3 Feature Extraction Infusion Network (FEIN)

The FiLM network initially used in the BC-Z approach [27] requires a task description and a human demonstration video as inputs. However, this approach isn't directly applicable to our specific case. Therefore, we designed a novel network architecture that establishes connections between the current audio-text inputs and the gestures observed in the previous time window. Our dual goals

were to ensure coherent gesture generation by conditioning on previous gestures and to inject additional contextual information into the current context.

To achieve these goals, we use three separate stacks of 1D convolutional layers to process the concatenated audio-text data and gesture information. This approach results in an embedding with an enriched spatial feature space, effectively capturing important spatial relationships. For meaningful interplay within these embeddings, a multi-head attention mechanism is incorporated. In this mechanism, the gesture embedding served as both query and value, while the audio-text embedding acts as the key. The goal of this attention-based embedding is to learn complex dependencies between gestures and audio-text data. The resulting attention-based embedding then traverses two different feed-forward networks. Each network consisted of two linear layers with SiLU activation functions to promote non-linearity and information propagation. A normalization layer completes each network, ensuring consistent and stable feature representations. This architectural configuration aims to facilitate the extraction of two essential feature networks: the $\gamma$-network and the $\beta$-network. These networks contain critical information for the following control model. Within the control network architecture, the role of the $\gamma$-network is to provide timing information about previous gestures to the embedding. This helps to maintain gesture consistency across time windows and counteract fragmented gestures. On the other hand, the $\beta$-network, due to its additive nature, provides nuanced details to the embedding. This feature allows the framework to capture subtle gestures that might be suppressed by the relatively coarse influence of the $\gamma$-network.

## 3.4 Control Network

The embedding network, derived from the transformer network, along with the $\gamma$ and $\beta$ networks from the FEIN model, serve as inputs for the control network. This network architecture is founded

**Table 1: The employed joints and their corresponding categorizations within the control network**

| Body part | number of joints | joints |
|---|---|---|
| **root** | 3 | b_root |
| **upper body** | 21 | b_spine0, b_spine1, b_spine2, b_spine3, b_neck0, b_head |
| **left leg** | 6 | b_l_upleg, b_l_leg |
| **right leg** | 6 | b_r_upleg, b_r_leg |
| **left arm** | 18 | b_l_shoulder, b_l_arm, b_l_arm_twist, b_l_forearm, b_l_wrist_twist, b_l_wrist |
| **left hand** | 48 | b_l_pinky1...3, b_l_ring1...3, b_l_middle1...3, b_l_index1...3, b_l_thumb0...3 |
| **right arm** | 18 | b_r_shoulder, b_r_arm, b_r_arm_twist, b_r_forearm, b_r_wrist_twist, b_r_wrist |
| **right hand** | 48 | b_r_thumb0...3, b_r_pinky1...3, b_r_middle1...3, b_r_ring1...3, b_r_index1...3 |

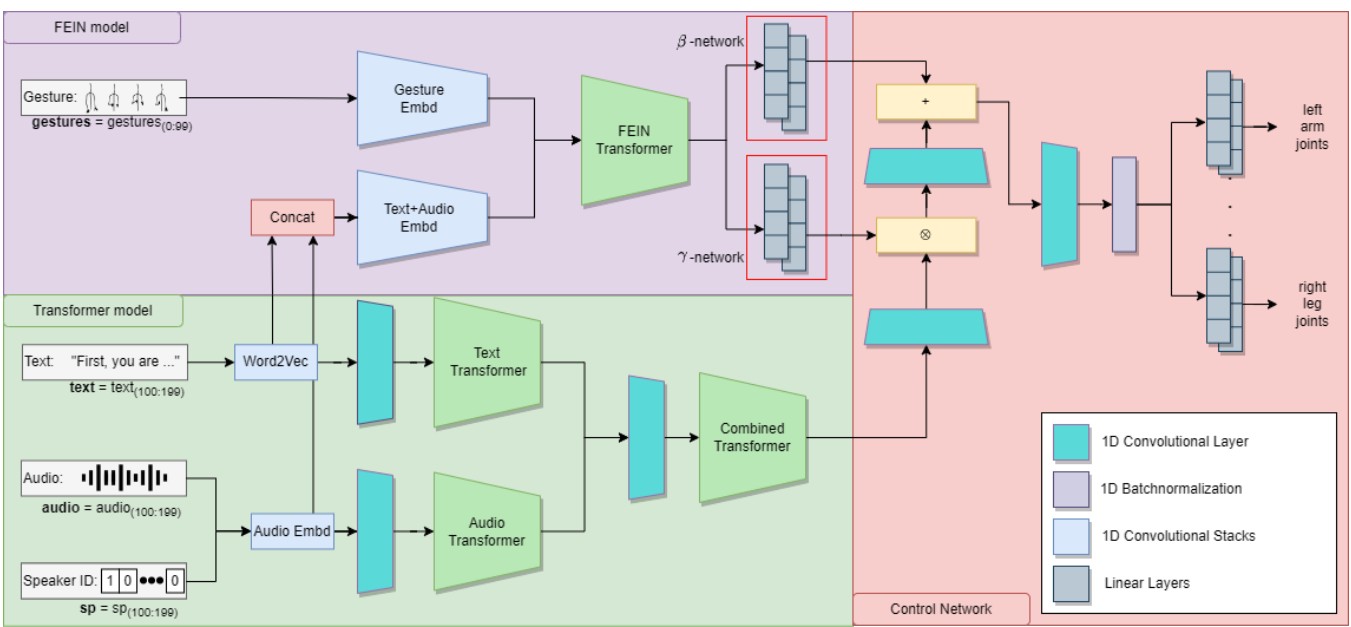

**Figure 1: Top: The proposed FEIN model with the convolutional embedder, transformer block, and $\gamma$- and $\beta$-FCN. Bottom: Transformer model with transformer blocks. Right: Control network with convolutional layers and $\gamma$ and $\beta$ infusion. All inputs (Gesture, Text, Audio, Speaker ID) consist of concatenated speaker and interlocutor information. The subscripts (0:99) and (100:199) denote distinct time windows represented by the input data.**

on the framework proposed by Jang et al. [27]. Initially, the embedding undergoes convolutional layer processing, resulting in a distilled embedding. Subsequently, this distilled embedding is enriched through element-wise multiplication with the $\gamma$-network output, which effectively integrates contextual information from the FEIN module. A subsequent convolutional layer processes the modulated output, combining information and yielding a transformed embedding. To further infuse the embedding with contextual cues, the transformed embedding is subject to element-wise addition with the $\beta$-network output. This step augments the embedding with supplementary contextual information. Following a final convolutional layer, the output is normalized, yielding a vector that merges current relevant features with essential contextual information. This integration is pivotal for generating coherent gestures, especially when considering the influence of preceding gestures. This processed vector then progresses through a sequence of fully connected networks (FCNs), with each FCN generating joint configurations for specific body parts, see Figure 1. This design imparts fine-grained control over individual body parts, thus facilitating precise manipulation of the model's movements. The employment of independent body-part-specific FCNs allows the framework to extract distinct features from the shared embedding, enabling a body-part-specific feature space.

### 3.5 Loss

The loss functions used in our framework are defined as follows. For the discriminator, the loss function is given by:

$$\mathcal{L}_{Dwdiv}(\mathbf{x}, D(\mathbf{z})) = Dis(\mathbf{x}) - Dis(D(\mathbf{z})) + \delta|\nabla_{\hat{\mathbf{x}}} Dis(\hat{\mathbf{x}})|^p \quad (4)$$

Here, $Dis$ represents the discriminator function, $\mathbf{x}$ represents the original dataset, and $\mathbf{z}$ represents the reconstructed data. The hyperparameter $\delta$ controls the magnitude of the divergence penalty. The first component of the loss, $Dis(\mathbf{x}) - Dis(D(\mathbf{z}))$, measures the dissimilarity between the real sample $\mathbf{x}$ and the output of our framework, $D(\mathbf{z})$. The second term, $\delta|\nabla_{\hat{\mathbf{x}}} Dis(\hat{\mathbf{x}})|^p$, corresponds to the divergence penalty, which encourages the generated sample $D(\mathbf{z})$ to closely resemble the distribution of real data. The generator loss function is defined as:

$$\mathcal{L}_{Gwdiv} = Dis(D(\mathbf{z})) \quad (5)$$

This loss function aims to minimize the output of the discriminator, specifically the evaluation of $Dis(D(\mathbf{z}))$.

For behavior cloning, we employ a scaled version of the smoothed L1 loss, defined as:

$$L1 = \begin{cases} \frac{0.5\theta(\frac{x}{\theta} - \frac{z}{\theta})^2}{\beta}, & \text{if } |x - z| < \beta \\ \theta|\frac{x}{\theta} - \frac{z}{\theta}| - 0.5\beta, & \text{otherwise} \end{cases} \quad (6)$$

This loss function is applied to the positions $y$ and $\hat{y}$, velocities $y'$ and $\hat{y}'$, and accelerations $y''$ and $\hat{y}''$. For this, the gradients are calculated using the following formula:

$$f(y) = \sum_{i=0}^{2} \lambda_i \frac{d^i y}{dt^i} \quad (7)$$

$$\mathcal{L}_{bc} = L1(f(y_i), f(\hat{y}_i)) \quad (8)$$

In these equations, $y$ represents the true gestures, while $\hat{y}$ denotes the predicted gestures. The function $f(y)$ calculates the gradients of the variable or function $y$ with respect to time. The superscript

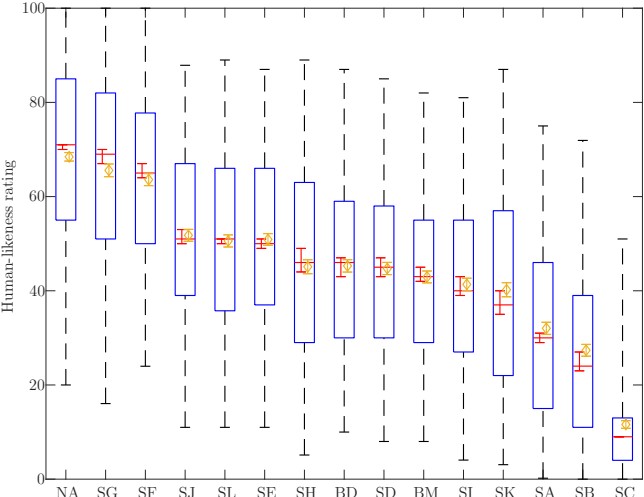

**Figure 2: Box plot visualization for the human-likeness study, provided by the GENEA Challenge 2023 [31]. Our framework is labeled SE. Median ratings are shown as red bars (with 0.05 CI) and mean ratings as yellow diamonds (with 0.05 CI). Box edges indicate the 25th and 75th percentiles. Whiskers cover 95% of ratings for each condition.**

$i$ in $\frac{d^i y}{dt^i}$ indicates the order of the derivative, ranging from 0 to 2. The $\lambda_i$ terms are scaling factors applied to the position, velocity, and acceleration losses.

The term $\mathcal{L}_{bc}$ corresponds to the loss function used for back-propagation. It is computed as the average of the individual loss terms $L_i$ over a dataset of size $N$. Each $L_i$ measures the dissimilarity between the calculated gradients $f(y_i)$ and the target gradients $f(y_i^*)$. Together, this loss ensures a temporal consistency of the generated gestures. The overall loss function used in our framework is a combination of the behavior cloning loss ($\mathcal{L}_{bc}$) and the discriminator loss ($\mathcal{L}_{G_{wdiv}}$):

$$\mathcal{L}_{total} = \mathcal{L}_{bc} + \mathbb{1}_n \cdot \lambda_g \mathcal{L}_{G_{wdiv}} \tag{9}$$

Here, $\mathbb{1}_n(s)$ is an indicator function defined as:

$$\mathbb{1}_n(s) = \begin{cases} 1, & \text{if } s \% n = 0 \\ 0, & \text{otherwise} \end{cases} \tag{10}$$

This indicator function is used to determine when to apply the discriminator loss. The parameter $n$ controls the frequency of applying the discriminator loss, and the scaling factor $\lambda_g$ adjusts the relative importance of the discriminator loss compared to the behavior cloning loss. By combining these components, the overall loss function guides the training process to improve the quality and consistency of the generated gestures.

## 3.6 Data Processing

The Genea Challenge 2023 provided an adapted version of the Talking With Hands 16.2M dataset [33], extended to a dyadic setting involving both a speaker and an interlocutor. This dataset encompasses various modalities, including 3D full-body gesture data,

audio data, text transcripts, and the speaker ID, all organized separately for the speaker and the interlocutor. As part of the challenge, the data was pre-separated into a training set of 371 sequences, a validation set of 40 sequences, and a test set of 69 sequences. Each sequence is approximately 1 minute in length, with a sample rate of 44100 Hz for the audio data. The gesture data was recorded at 30 frames per second. Since the challenge required the generation of the speaker for the test set, this data was omitted.

For our approach, we built upon the preprocessing pipeline established by Chang et al. [11], making necessary modifications to suit our specific requirements. For the audio data, we used multiple feature extraction techniques to obtain three different features: Mel Frequency Cepstral Coefficients (MFCC) with 40 dimensions, Mel Spectrograms with 64 filter banks, and prosody features. All audio features were computed using a window length of 4096 and a hop length of 1470. Regarding the text transcripts, we used the FastText word embedding model [7], which assigns a 300-dimensional vector representation to each word in the transcript. Since the temporal duration of each word is known, we generated a vector of size [sequence length, 300] containing the corresponding word embedding vector for each word's duration. For the gesture data, we transformed the rotation of each body and finger joint in the BVH file into an exponential map representation [22]. This transformation resulted in 56 3D body joints for the gesture data.

In the post-processing phase of the gesture output, we performed two operations. First, we clipped the angle of each generated body joint to be within the range of the 2nd and 98th percentiles of the corresponding joint in the training data. This clipping step ensured that the generated angles remained within a reasonable range. Afterward, we applied a rolling window calculation over 50 frames to smooth the generated output and improve its temporal coherence.

## 3.7 Training procedure

The training procedure incorporates both behavior cloning and the WGAN architecture. In our setup, the network is responsible for generating gestures, while the discriminator is used to discriminate between the generated data and the original data. We chose a batch size of 128 and a sequence length of 200 frames, which corresponds to two frame windows: $t_{-1} := [0 - 99]$ and $t_0 := [100 - 199]$. For the optimizer, we use AdamW [37] with a weight decay parameter of 0.01 for both the FEIN network and the discriminator. For the FEIN model, we select a learning rate of $5e - 5$, while the discriminator utilizes a learning rate of $1e - 4$. During training, we set the scaling factor $\lambda_g$ to 0.05.

The audio and text data used in training comes from $t_0$, while the gesture data is sourced from $t_{-1}$. After each prediction step, we optimize the model using the loss function described in 9, and we optimize the discriminator accordingly using its loss function, as defined in 4. To prevent the network from consistently outperforming the discriminator and to stabilize the training, we apply the 5 loss only every $n = 4$ steps. In total, we trained our framework for 60 epochs. Every 10 epochs, we computed the validation loss and used the best-performing model to generate the evaluation data.

# 4 EVALUATION

During the training phase of the framework, we conducted a thorough analysis of various framework configurations, experimenting with different numbers of transformer blocks and parameters. We also explored frameworks that generated gestures for both the main agent and the interlocutor, as well as different input data for the FEIN model. Among these tested frameworks, many did not yield satisfactory results in terms of generating realistic and coherent gestures. As a result, we selected the framework proposed in this study as the most suitable for our purposes.

The main evaluation of the framework was performed alongside other approaches within the GENEA Challenge 2023. Since the evaluation of generated co-speech gestures is largely subjective and objective measures that strongly correlate with subjective evaluations are lacking [41], the evaluation focused primarily on subjective measures. Three specific aspects were evaluated: "Human-Likeness", "Appropriateness for Agent Speech", and "Appropriateness for the Interlocutor". To ensure anonymity, all published results were anonymized and assigned unique labels. Our framework was labeled *SE*.

## 4.1 Human-Likeness

The results of the Human-Likeness evaluation are shown in Figure 2, illustrating the rating distribution obtained for the different approaches. Figure 3 highlights the significant differences between the competitors. Here, our framework receives significantly higher ratings than the dyadic baseline (**BD**), the monadic baseline (**BM**), as well as the approaches **SH**, **SD**, **SI**, **SK**, **SA**, **SB**, and **SC**. On

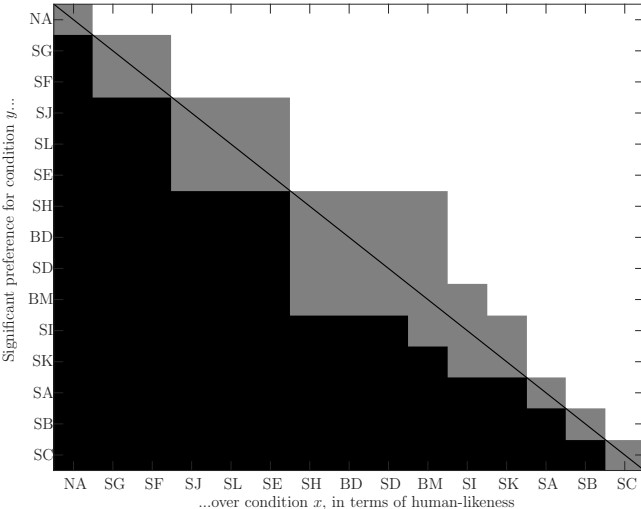

**Figure 3: Significant differences between all approaches, provided by GENEA Challenge 2023 [31]. Our framework is labeled SE. White indicates that the condition on the y-axis is rated significantly higher than the one on the x-axis, while black indicates the opposite (y-rated below x). Gray indicates no statistically significant difference at a significance level of $\alpha = 0.05$, after applying the Holm-Bonferroni correction.**

the other hand, compared to the natural motion (**NA**) and the approaches **SG** and **SF**, our framework receives significantly lower ratings for human-likeness. There were no significant differences in terms of human-likeness between our approach and the approaches **SJ** and **SL**.

A significant limitation of our approach, especially concerning human-like gesturing, was the lack of finger movement in all of the generated gestures. Although we trained our framework to produce output for the finger bones, the resulting gestures consistently exhibited a static finger position. Any changes observed in the finger bones were primarily intended to prevent the introduction of artifacts, rather than to add meaningful information to the generated gestures.

Another notable issue was the rapid change of poses in our framework. Although the evaluation only captured footage from the knees up, to prevent any foot sliding from influencing the evaluation, our model consistently exhibited movements that involved a redistribution of weight in the lower part of the torso. Such movements may have compromised the naturalness of the generated gestures and led to a lower ranking in the human-likeness evaluation.

## 4.2 Appropriateness

The results of the speech appropriateness evaluation for the main agent are depicted in Figure 4a. These ratings indicate the likelihood of each framework being preferred with matching or mismatching gestures. Our proposed framework, labeled **SE**, demonstrates statistical significance in terms of speech appropriateness compared to random chance. However, it is notably inferior to framework **SG**, which exhibits significantly better performance. Additionally, there is no significant difference between our framework and the approaches **SJ**, **SF**, **SK**, **SD**, **SI**, **SK**, **SB**, **SA**, and **SH** in terms of speech appropriateness. The results of the appropriateness of gestures in response to the interlocutor are presented in Figure 4b. These ratings reflect the likelihood of each framework being preferred with matching or mismatching gestures. Our framework does not exhibit statistical significance compared to random chance in this aspect. Our model does achieve a significantly higher mean appropriateness score (MAS) compared to frameworks **SG** and **SH**, and a significantly lower MAS compared to the natural motion **NA**. Furthermore, our model does not differ significantly from the dyadic and monadic baselines, as well as frameworks **SA**, **SB**, **SL**, **SF**, **SI**, **SD**, **SJ**, **SC**, and **SK**, in terms of appropriateness of gestures in response to the interlocutor.

The evaluation results presented here show a notable discrepancy when compared to the results of the human similarity evaluation. While our framework is able to generate co-speech gestures that are perceived as more human-like than the baseline used in the challenge, this does not mean that the generated gestures are perceived as more appropriate for the given context than the baseline. Although the lack of finger bone information could be a possible explanation for this, we suggest that it is indicative of a general problem common to all current approaches to co-speech gesture generation. Current approaches excel at producing gestures that appear natural and unobtrusive within a given conversation, which is already a commendable achievement for human-agent interaction.

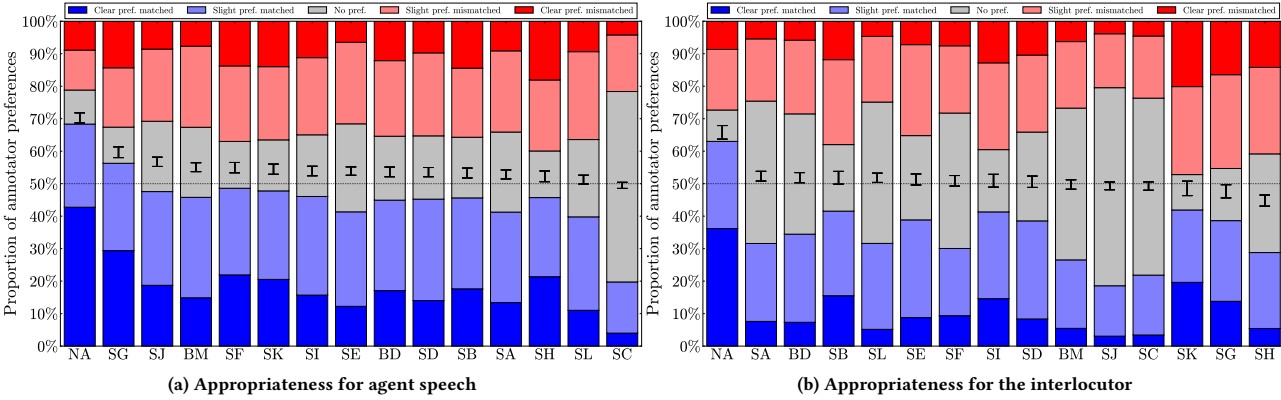

(a) Appropriateness for agent speech                                    (b) Appropriateness for the interlocutor

**Figure 4: Bar plots visualizing the response distribution in the appropriateness studies, provided by the GENEA Challenge 2023 [31]. Our framework is labeled SE. The blue bar (bottom) represents responses where subjects preferred the matched motion, the light grey bar (middle) represents tied responses, and the red bar (top) represents responses preferring mismatched motion, with the height of each bar being proportional to the fraction of responses in each category. Lighter colors correspond to slight preference, and darker colors to clear preference. On top of each bar is also a confidence interval for the mean appropriateness score, scaled to fit the current axes. The dotted black line indicates chance-level performance. Conditions are ordered by mean appropriateness score.**

However, this still falls well short of replicating human-to-human interaction. In human-to-human communication, individuals convey additional meaning through their gestures [14], which is based on a shared mental model of the current conversation, themselves, and the conversation partner [15, 16]. With this shared understanding, conversational partners can adapt their gestures to each other and effectively convey meaningful information. Since our framework, and to the best of our knowledge all other available co-speech gesture approaches, lacks this essential insight into the conversation partner, the generated gestures appear highly interchangeable to any human evaluator.

**Table 2: The Fréchet Gesture Distance (FGD) distance for each ablation modification, calculated both in the feature space (FGD F-space) and the raw data space (FGD R-space). For both distances, lower is better.**

| Methods | FGD F-space ↓ | FGD R-space ↓ |
|---|---|---|
| **natural motion** | 0.00 | 0.00 |
| **w/o transformer** | 169.93 | 3334.14 |
| **w/o $\gamma$-network** | 84.45 | 2667.33 |
| **w/o $\beta$-network** | 61.76 | 1879.82 |
| **w/o audio** | 50.93 | 965.05 |
| **w/o text** | 43.9 | 1099.48 |
| **w/o main audio** | 34.98 | 758.62 |
| **w/o inter text** | 31.26 | 767.28 |
| **w/o main text** | 29.49 | 777.91 |
| **w/o inter audio** | 28.54 | 680.66 |
| **original** | 23.03 | 533.04 |

## 5  ABLATION STUDY

In order to assess the specific contributions of each component within our proposed framework, we conducted an ablation study. First, different input configurations were investigated, including the exclusion of all textual input ("w/o text"), the exclusion of all audio input ("w/o audio"), and the selective removal of these modalities for the main speaker ("w/o main audio" and "w/o main text") as well as for the interlocutor ("w/o inter audio" and "w/o inter text"). Furthermore, different architectural configurations were explored, including deactivation of the output of the combined transformer ("w/o transformer"), deactivation of the $\beta$-network ("w/o $\beta$-network"), and exclusion of the multiplication process involving the $\gamma$-network (referred to as "w/o $\gamma$-network"). The distinction in the generated gestures was measured by using the Fréchet Gesture Distance (FGD), as defined by Yoon et al. [55], for each modification. The evaluation of this distance was performed both in the feature space of the autoencoder network given by the GENEA 2023 challenge and in the context of the raw data space, similar to Ahuja et al. [2]. Detailed results are presented in Table 2. We make an example video of all modifications available online[1].

As can be expected, each modification of the framework leads to an increase in the FGD, both in the feature space and in the raw data space. In terms of the modality-specific inputs associated with the interactive partner, all modifications lead to a comparable increase in the FGD. In particular, the removal of the interlocutor's audio produced the smallest change, while the exclusion of the main speaker's audio produced the largest change. The complete removal of both textual and audio information led to a sharp increase in FGD. Visual inspection of the generated gestures revealed instances of elaborate but misaligned gestures in cases of audio removal,

---

[1] https://vimeo.com/853326587

whereas small and infrequent gestures were observed following text removal.

Looking at the modifications of the architectural configurations, it becomes clear that the transformer model has successfully learned to generate the gestures since the removal leads to strongly degraded performance and the largest increase in FGD of all modifications. Similarly, the removal of the $\beta$ network and the $\gamma$ network leads to a deterioration of the performance. Looking at the visual results of the $\beta$ network, the gestures still show a natural fluid movement but are mainly concentrated in front of the chest and do not show any obvious finger movement. On the other hand, the visual results from the $\gamma$ network show fast, erratic movements of the hands and upper body, with some unnatural poses. These results support our intended design choices, with the $\gamma$-network focusing mainly on smoothing the temporal information of the generated gestures, while the $\beta$-network refines the generated gestures to allow for more elaborate hand movements.

## 6 CONCLUSION

Our framework presents a novel approach to co-speech gesture generation inspired by robotic imitation learning and based on a behavior cloning architecture. We combine a transformer architecture with a generative adversarial network to create a model that ranks in the top half of the GENEA Challenge 2023 [31]. Although the model did not achieve results comparable to natural motion, we believe that additional training time and more sophisticated input segmentation could lead to improved results. An effective strategy may involve the use of only historical data in the FEIN model to ensure that the input data consists only of aligned gesture, audio, and text data. In addition, the use of a finer-grained control network that distinguishes separate body parts, such as hands and arms, could have the potential to improve the generated gestures. Increasing the feedback provided by the discriminator model in later stages of training is another way to improve performance, as the discriminator shows diminishing returns as training progresses. Additionally, selectively freezing certain models within our framework during later stages of training to focus on refining gestures could lead to performance improvements. Similarly, exploring alternative inference methods, such as predicting one frame at a time or adjusting the time window, may also help to improve the capabilities of the framework. In conclusion, we believe that our architecture demonstrates the potential to generate gestures that exhibit some human-like characteristics, and we believe that there are several ways in which our framework could be improved in the future. Finally, we hypothesize that the integration of frameworks introduced in multimodal robot learning could further enhance the performance of future gesture generation models.

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
