# OpenReview forum: "FEIN-Z: Autoregressive Behavior Cloning for Speech-Driven Gesture Generation"
_ACM.org/ICMI/2023/Workshop/GENEA_Challenge — GENEA Challenge 2023 Mainproceeding_

### Official Review · Reviewer_2KcF · 2023-08-01
**Interesting adaptation of imitation learning approach**

**Rating:** 6
**Confidence:** 4

**Review:**

This paper proposes a novel model based on the BC-Z imitation learning model with several architectural improvements and modifications for gesture generation. The proposed approach is sound. It is interesting that the authors are motivated by imitation learning, which have not yet been actively applied in the gesture generation field.

Comments and questions:
- While adopting an imitation learning approach is an interesting direction, the authors should elaborate some more to explain why imitation learning is beneficial in gesture generation. For example, what components of the proposed method contribute to “generate elaborate and complex outputs by acquiring knowledge from a relatively limited data set” (lines 188-190)?
- Some explanations of the proposed method need to be included.
    * What network architecture is used as the discriminator?
    * In Figure 1, what are the number in parenthesis, e.g., (0:100), (100:200)?
    * Why did you use a higher learning rate for the discriminator?
    * Lines 574-576: “Among these tested frameworks, ... coherent gestures.” Is it possible to provide some analysis (quantitative or qualitative) on this aspect? How about visualizing gestures, e.g., without WGAN, FEIN, etc.?

- Typo:
    * Figure 1: Word2Vec -> fastText
    * Line 389: Fort his -> For this
    * Line 396: y* -> \hat{y}?

---

### Official Review · Reviewer_87G5 · 2023-08-01
**An interesting paper**

**Rating:** 8
**Confidence:** 5

**Review:**

[Paper Summary]

This paper proposed a co-speech generation framework combining transformer and Wasserstein GAN architecture. The model was inspired by BC-Z network architecture, and the authors replaced the vision network component in BC-Z with an attention-based network.  The model takes audio, text, and speaker identity information from both the main agent and the interlocutor as inputs, alongside gestures from the interlocutor.
The model was trained and tested on the Talking with Hands dataset. Based on the network performance, which was verified in terms of Human-Likeness, Appropriateness for agent speech, and Appropriateness for the interlocutor, the authors claimed that the solution showed good results and significant improvements in human-likeness over the GENEA baseline.

[Strengths]

1. This is a well-investigated work, and the proposed approach seems to be interesting. It is nice to see the BC-Z network, designed originally for the robot imitation learning task, has been modified and adapted to the co-speech gesture generation domain.

2. The paper is also well-written and organized. The proposed approach has been explained in a careful manner.

3. Qualitative discussions have been made to reason about the low quality of the generated motions compared to related approaches in the GENEA 2023 challenge.

[Weakness]

There are several points that the authors may consider in the revised version:

1. Better clarification of the designed approach. What is the purpose of designing a $\beta$ and $\gamma$ network? In other words, which features did the authors aim to extract from that two individual networks?

2. Objective evaluation. If the position, velocity, and acceleration loss are implemented in the loss function. Conducting an ablation study to highlight the contribution of those losses to the total loss function L_total would be interesting. In this case, the authors can rely on objective metrics (that measure the errors of position, velocity, and acceleration between generated and GT motions) introduced in the previous GENEA challenges.

3. Qualitative results. For instance, a figure of generated motion can be included to support discussion about the lack of finger movement that the authors mentioned in section 4.1.

4. Typos. The paper should be proofread again to remove typos, for instance, L.389 on Page4.

---

### Official Review · Reviewer_JLAL · 2023-08-02
**A new model to generate different joints**

**Rating:** 6
**Confidence:** 4

**Review:**

Paper Summary:
The paper presents a new model that employs modules such as BC-Z (Control Network) and FEIN to generate different joints. However, the paper does not provide detailed explanations about the specific roles and effects of these modules, nor does it conduct ablation experiments to demonstrate the effectiveness of each module.

Relevance:
The topic of the paper is highly relevant to the conference theme, exploring the application of artificial intelligence in generating specific joints.

Significance:
The research in this paper is of great importance for understanding and improving joint generation models. However, due to the lack of sufficient empirical evidence, the significance of its contribution needs further verification.

Paper Strengths:
The paper proposes a new model, attempting to generate different joints.
The paper provides detailed descriptions of the design and implementation of the model.

Paper Weaknesses:
The model in the paper seems to be able to generate directly in one step, without the need to generate different joints, which questions the appropriateness of its being called a "control model".
The paper does not provide detailed explanations about the specific roles and effects of modules like BC-Z (Control Network) and FEIN, nor does it conduct ablation experiments to demonstrate the effectiveness of each module.
The post-processing (trimming joints) part in the paper is not well explained, and it is unclear whether it is because the generated results do not meet expectations.
There are some issues in the evaluation part of the paper, such as contradictions in the descriptions at L623 and L626.
There are some issues with the writing of the paper, such as unreasonable layout, lack of equation numbers, inconsistent fonts, and some expressions that are confusing.

Further Comments:
The specific joints generated in Figure 1 should be listed, the original BC-Z utilizes no use of loss (Huber loss, log loss) for XYZ, rotation, Gripper, and this work is just generating for different joints, I don't know if it works here, it seems like it can be generated directly in one step, calling it a control model is a bit inappropriate Why post-processing (pruning the joints), is it because the generated results are not as expected? The model should be able to learn on its own that the data should lie between the distributions of the training data. The modules used are BC-Z (Control Network), FEIN, etc. I would like to know if each of these modules is useful and which one has the greatest impact on the results, it would be useful if you could add ablation experiments. Please check that your assessment is written correctly, L623 is correct, but L626 is again written as not significantly different from SF, which is clearly evident from Figures 2 and 3. Does the method use hand joints? L506 writes hand gestures, but does L628 mean that hand movements cannot be generated correctly? Writing problem: Why is there a large blank space at the bottom left of the first page? Suggest adjusting the layout; the formula near L247 is not labeled, as well as MultiHead and Swish fonts are not the same; L520, lowercase T after comma, as well as the puzzling phrase "The network is responsible for generating gestures. "This is very confusing, and the way it is written in many places could be optimized; in Figure 1, the frames are 1:100 and 100:200, and in L552 they are 0-99 and 100-199, which I would suggest to keep the same; in L560, the 6 should be changed to Equation (6), and the 2 in L 563 is the same. And some descriptions in the paper need further clarification, such as "What does 'a redistribution of weight in the lower part of the torso' in L663 mean?".

---

### Decision · Program_Chairs · 2023-08-04

**Decision:**

Accept (Main proceeding)

**Comment:**

All the reviewers favoured accepting this paper with relatively high scores of 6,8,6. The chairs agree to accept this paper to the Main ICMI Proceedings. Please read the reviews below carefully and revise the paper for the camera-ready version. The main changes requested are the following:
1. Provide better clarification of the designed approach as requested by several reviewers.
2. Provide detailed explanations about the specific roles and effects of modules like BC-Z and FEIN.
3. Provide qualitative results, for example, by linking to a video.
4. Proofread again to remove typos.
5. Add missing details of the proposed method, as three reviewers pointed out.
6. Consider adding ablation studies if you have time to do.